# Can the COVID-19 Pandemic Improve the Management of Solid Organ Transplant Recipients?

**DOI:** 10.3390/v14091860

**Published:** 2022-08-24

**Authors:** Arnaud Del Bello, Olivier Marion, Jacques Izopet, Nassim Kamar

**Affiliations:** 1Department of Nephrology and Organ Transplantation, Toulouse Rangueil University Hospital, 31059 Toulouse, France; 2Toulouse Institute for Infectious and Inflammatory Diseases (Infinity), INSERM UMR 1291, 31300 Toulouse, France; 3University Toulouse III—Paul Sabatier, 31000 Toulouse, France; 4Laboratory of Virology, Toulouse Purpan University Hospital, 31300 Toulouse, France

**Keywords:** COVID-19, organ transplantation, monoclonal antibodies, prevention, omicron

## Abstract

Increased mortality due to SARS-CoV-2 infection was observed among solid organ transplant patients. During the pandemic, in order to prevent and treat COVID-19 infections in this context, several innovative procedures and therapies were initiated within a short period of time. A large number of these innovations can be applied and expanded to improve the management of non-COVID-19 infectious diseases in solid organ transplant patients and in the case of a future pandemic. In this vein, the present paper reviews and discusses medical care system adaptation, modification of immunosuppression, adjuvant innovative therapies, the role of laboratory expertise, and the prevention of infections as examples of such innovations.

## 1. Introduction

Very soon after the start of the COVID-19 pandemic at the end of 2019 and the beginning of 2020, high mortality (up to 30%) was reported among solid organ transplant (SOT) recipients [1,2,3,4], especially in areas with high viral circulation [5]. Later on, it was confirmed that this population was at high risk of symptomatic severe acute respiratory syndrome coronavirus-2 (SARS-CoV-2) infection and severe COVID-19. This is the result of an increased rate of comorbidities (such as advanced age, cardiovascular diseases, diabetes mellitus, being overweight, and having impaired kidney function) observed in this population, as well as of decreased immune responses due to the immunosuppressants administered to prevent acute rejection [6]. In addition, nosocomial transmissions were observed in these patients, who often attend health care structures for routine follow-up to receive specific therapies (such as belatacept infusions) or to be treated for complications [7].

In the face of this pandemic, health care workers and researchers rapidly concentrated their efforts on reorganizing the medical care system, finding efficient treatments against severe COVID-19, and developing and implementing prevention measures, such as the vaccination of large cohorts of immunosuppressed patients. Within this short period, the number of publications focusing on COVID-19 increased dramatically, while there was a substantial decrease in non-COVID-19-related research [8].

A considerable proportion of the innovative procedures and therapies that were introduced during the pandemic can be applied and expanded to improve the management of non-COVID-19 infectious diseases in SOT patients, and in the case of a future pandemic. The present paper briefly discusses some of these measures.

## 2. Modification of the Medical Care System

The lockdowns enforced in several countries prompted transplant physicians to modify the management of SOT patients. Hence, for instance, telemedicine developed rapidly and teleconsultation was initiated to assure follow-up for transplant patients. The use of mobile phone applications increased dramatically. Due to the risk of SARS-CoV-2 nosocomial transmission, it was recommended that clinicians avoid admitting patients to hospitals to receive specific therapies. For instance, initially, transplant physicians questioned the in-hospital administration of intravenous belatacept, and suggested replacing it with oral immunosuppressive regimens that required frequent laboratory testing [9]. Since the clinical and biological consequences were unpredictable, we organized a specific infection control protocol for organ transplant patients receiving maintenance belatacept therapy [10]. All patients were screened over the phone for respiratory and/or gastrointestinal symptoms the day before the scheduled infusion. Patients who had a fever, respiratory symptoms, gastrointestinal disorders, or who had been in contact with a person who was SARS-CoV-2-positive were directed to a unit dedicated to suspected cases of COVID-19. A different isolated section in the outpatient unit was dedicated to patients without symptoms of COVID-19. In addition, at admission, patients were interviewed to detect alarming symptoms. Since the nosocomial transmissions of influenza viruses and *Pneumocystis Jirovecii* were reported in this population [11], a similar protocol could at least be implemented while influenza viruses were in circulation. Furthermore, as is current practice, the systematic wearing of masks can be highly recommended in SOT patients admitted to in- or out-patient clinics. Later on, during the pandemic, in France, trained nurses were permitted to administer belatacept during home visits. This practice is still allowed and is permitted to reduce the number of hospitalizations.

In order to continue to pursue SOT transplantation during the pandemic, centers had to specify a “COVID-free” hospitalization ward. Before donation and transplantation, patients were tested for SARS-CoV-2, and a chest CT-Scan was very often performed. It was recommended that de novo transplant patients and/or SARS-CoV-2-negative patients be hospitalized in a dedicated unit, while SARS-CoV-2-positive patients were isolated in another unit. As was the case during the pandemic, it seems logical to group infected patients, for instance, those with influenza viruses or *Pneumocystis Jirovecii*, in a dedicated unit, or at least in a dedicated part of a unit. A dedicated nurse team could take care of these infected patients. This would probably decrease the risk of nosocomial transmission. Finally, SARS-CoV-2-infected patients were systematically flagged in the hospital information system. Identifying patients infected by other viruses or by multi-drug-resistant bacteria in the hospital information system will also allow clinicians to isolate patients at admission and to reduce the risk of transmission between SOT patients. Infection by multi-drug-resistant bacteria is associated with a high risk of mortality among SOT patients [12].

## 3. Management of Immunosuppression during Severe Infectious Disease Complications

At present, there is still no recommendation for the management of immunosuppression in SOT recipients presenting severe infections. This was also the case for COVID-19 [13]. In patients infected with SARS-CoV-2, there was, on the one hand, the risk of evolution to severe acute respiratory distress, and on the other hand, the risk of acute rejection, especially in non-kidney-transplant patients [1,14,15]. In addition, it has been suggested that pursuing immunosuppressants could have a beneficial effect on hyperinflammation in ARDS [15]. Finally, some antiviral drugs that have been used to date, such as Remdesivir or Paxlovid^®^ (nirmatrelvir/ritonavir), have a drug–drug interaction with immunosuppressants, leading to increased dose exposure [16].

Registry studies have shown that the administration of anti-metabolites was stopped in a large proportion of in-patients. In addition, the use of calcineurin and mammalian target of rapamycin inhibitors was decreased in a significant proportion of patients [3,17,18]. Conversely, the use of steroids was increased in most patients. In a cohort of 47 kidney transplant patients hospitalized for COVID-19, in whom immunosuppression, mainly antimetabolites, was reduced (83%), no acute rejection or de novo DSAs was observed until 3 months after discharge [19]. In contrast, in a larger cohort (179 kidney transplant patients infected with SARS-CoV-2), the incidence of de novo DSA after the reduction of immunosuppression was at 4% [20]. Cases of antibody-mediated rejection were also reported [21]. This is in line with previous studies that showed a risk of development of de novo DSAs in SOT patients admitted to intensive care units and who had immunosuppression withdrawal [22,23].

Regarding out-patients, the management of immunosuppression varied between centers. No robust data were published regarding the management of immunosuppression in this setting. For instance, in our center, immunosuppression was not modified in patients not requiring hospitalization.

In summary, in the short term, the reduction does not seem to be harmful. Nevertheless, the mid- and long-term impacts, especially when DSAs occur, are still unknown. There is an urgent need to determine the optimal management of immunosuppression in SOT patients with severe infection.

## 4. Adjuvant Therapies for SOT Patients

### 4.1. Tocilizumab

A rapid increase and prolonged SARS-CoV-2 viral load, followed by an excessive and prolonged innate immune activation, was observed during COVID-19. In severe forms, a dysregulated immune response with a prothrombotic state, endothelial activation, and excessive pulmonary neutrophil recruitment emerges, with the formation of neutrophil extracellular traps. This leads to respiratory failure from acute respiratory distress syndrome and death. Notably, the Interferon type I and Interferon type III responses are reduced in COVID-19, reducing the early host innate activity. Moreover, high pro-inflammatory cytokine levels (IL-6, IL-8, IL-10, TNF-α, CRP, and IL-2R), the so-called “cytokine storm syndrome”, are observed in patients that develop severe forms of COVID-19 [24]. A late adaptative immune cell response, with suppressed Th1 adaptative antiviral immune response and CD8 and NK cell exhaustion or dysfunction, is observed in severe forms [25,26].

Small series [27] and, later on, randomized control trials [28,29] showed a lower risk of death and less need for mechanical ventilation in non-transplant patients given tocilizumab, an anti-IL-6 receptor blocker, as an adjuvant therapy in case of moderate or severe COVID-19. According to the NIH guidelines [30], anti-IL-6 receptor blockers are used with caution in SOT patients for treating chronic antibody-mediated rejection [31,32] or in desensitization protocols [33]. Physicians were reluctant to use tocilizumab in SOT patients. However, despite the use of prolonged immunosuppressive therapy, “cytokine storm syndrome” was observed in SOT recipients. In addition, similar to immunocompetent patients, tocilizumab seems to have a beneficial effect in treating SARS-CoV-2-infected SOT patients. In a single-center, retrospective, matched controlled study, Pereira et al. reported similar outcomes in 29 SOT patients, regardless of whether they were given tocilizumab for COVID-19 [1]. No safety issues were observed. The lack of demonstration of a beneficial effect of tocilizumab on mortality is probably related to the higher severity in patients given anti-IL6R blockers. In a multicenter retrospective study that included 80 kidney transplant recipients who have received tocilizumab, Perez-Saez et al. reported a beneficial effect of tocilizumab for controlling “cytokine storm syndrome” in cases of steroid treatment failure [34]. Additionally, we and others have reported cases of kidney transplant patients successfully being treated with tocilizumab for hemophagocytic syndrome related to COVID-19 [35,36]. Hence, this observation suggests that innovative therapies should also be proposed and tested in immunocompromised patients.

### 4.2. Anti-SARS-CoV-2 Neutralizing Monoclonal Antibodies

Several studies have shown the beneficial effect of anti-SARS-CoV-2 on neutralizing monoclonal antibodies when administered in the early period of infection and to in-patients not requiring oxygen with a decreasing viral load, COVID-19-related hospitalization, and death, particularly in high-risk populations [37,38]. Due to the risk of the emergence of drug-resistant variants [39], neutralizing monoclonal antibodies were not widely used in immunocompromised patients. However, interestingly, despite the lack of data for transplant patients, monoclonal antibodies were used in a very large number of SOT patients. Dhand et al. reported encouraging results in SOT patients given bamlanivimab monotherapy or casirivimab/imdevimab [40,41]. We compared the outcomes for 16 SOT patients who were given neutralizing monoclonal antibodies (bamlanivimab monotherapy, bamlanivimab/etesevimab, or casirivimab/imdevimab) early after infection with those of 32 SOT patients who did not receive this therapy. The proportions of patients who developed severe disease requiring ICU and of patients who died were significantly lower in patients who were offered monoclonal antibodies [29]. These data were recently confirmed in a large retrospective multicenter study [42]. Interestingly, despite the emergence of resistant variants in some patients who were given bamlanivimab/etesevimab, the use of casirivimab/imdevimab was not associated with the selection of resistant variants [43,44]. Conversely, the use of sotrovimab in combatting the Omicron variant was associated with selected resistant variants in half of the treated patients [45]. Therefore, despite the risk of selecting resistant variants, the use of monoclonal antibodies was beneficial for avoiding severe disease. However, virological parameters should be monitored closely and measures to limit virus spread should be reinforced. In immunocompromised patients with severe COVID-19, the use of convalescent plasma therapies was also found to be beneficial [46]. Hence, both monoclonal antibodies and convalescent plasma therapies were successfully used in transplant patients without having preliminary data in this setting. This suggests that, in case of another pandemic, new therapeutics should be rapidly tested in SOT patients and could be used in this specific population.

### 4.3. Virology Laboratory Expertise

The SARS-CoV-2 pandemic prompted the quick development of virological tools that were essential for diagnosis, treatment, and prophylaxis [47,48].

Beyond nucleic acid and antigen testing in nasopharyngeal swabs for detecting SARS-CoV-2, the characterization of the virus by whole-genome sequencing enabled genomic surveillance at a global level. New variants with distinct characteristics in terms of transmissibility, virulence, and immune escape were identified over time, preceding the different epidemic waves. Genomic data indicated no significant difference in the distribution of SARS-CoV-2 variants between solid organ transplant recipients and immunocompetent individuals. The characterization of SARS-CoV-2 quasispecies using third-generation long-fragment sequencing was also an effective means to analyze the selective pressure induced by the immune response or the administration of monoclonal antibodies at an individual level [49]. This approach allowed the quick identification of resistant variants in solid organ transplant recipients who were given strovimab [50] or tixagevimab/cilgavimab [51]. This close virological monitoring minimized the risk of the transmission of resistant variants to patients’ relatives and in this population with prolonged viral replication. The quantification of SARS-CoV-2 RNA using digital PCR was also very helpful for deciphering pathogenesis and analyzing the intrinsic antiviral activity of neutralizing monoclonal antibodies [43,45].

SARS-CoV-2 cultures under BSL-3 laboratory conditions were key to isolating new variants providing biological material for internal and external quality control and the development of serological assays designed to measure neutralizing antibodies. These assays were used for evaluating the performance of commercial serological tests [52,53] and for determining correlates of protection against SARS-CoV-2 [54,55,56].

Using validated serological assays that can measure neutralizing and binding antibodies, we were able to compare the humoral immune response post-vaccination between SOT recipients and healthcare workers [57]. Monitoring the antibody response, after primary vaccination and booster injections, in SOT recipients, as in other immunocompromised persons, was key for optimizing their management, including the decision to administer an additional vaccine booster or initiate pre-exposition prophylaxis using neutralizing monoclonal antibody therapy effective on circulating SARS-CoV-2 variants. Lastly, ELISpot assays and T-cell interferon-gamma release assays using spike recombinant proteins were conducted to analyze the T-cell response in vaccinated SOT recipients [58,59,60].

## 5. Prevention

### 5.1. Anti-SARS-CoV2 Vaccination

Within a very short period of time, anti-SARS-CoV-2 vaccines were developed. The vaccines most frequently deployed were those based on innovative technology using nucleoside-modified mRNA molecules encoding the SARS-CoV-2 spike protein [61]. Because of the increased mortality rates of COVID-19 in immunosuppressed patients and the lack of successful treatments, it was very quickly recommended that the vaccine be offered to immunosuppressed patients, including SOT recipients [62]. Similar to monoclonal antibodies, anti-SARS-CoV-2 vaccines were approved for SOT patients without appropriate studies in this population that is expected to have a low humoral response. Hence, when the vaccine became available, the first challenge was to contact all patients and organize the vaccination campaign. Transplant centers used different tools to contact their patients, e.g., emails, text messages, phone calls, social media, etc. This prompted all centers to create a complete database and a simple tool for contacting all patients. This tool is now available so that information pertaining to transplant patients can be procured rapidly. Secondly, we needed to organize the process of vaccinating patients. In our hospital, it was decided that the vaccine should be administered in a dedicated place within the hospital. In addition, as recommended by the Francophone Society for Transplantation, patients were asked to control their anti-SARS-CoV-2 serology after vaccination. Based on the serology results, we initially administered two doses [63], and thereafter three doses [64,65] to all patients. In weak and non-responding patients, a fourth dose was given [58]. However, this did not confer sufficient protection [58,66]. The levels conferring protection against infection were determined in studies performed on health care workers [56,67]. They varied over time according to the variants of concern. In addition, since data from all vaccinated patients were collected prospectively, several groups were able to identify risk factors for non-response to vaccination (summarized in [68]). Hence, the use of two major immunosuppressants, belatacept and mycophenolic acid, was associated with a worse humoral response to vaccination. This prompted some groups to modify immunosuppression to improve the humoral and cellular responses. The lessons from this large cohort vaccination campaign are threefold. Firstly, transplant centers informed their patients, developed tools to contact all their patients rapidly, organized vaccination, and collected data very rapidly. Secondly, COVID-19 raised the interest of transplant physicians in proposing vaccines to patients, identifying modifiable causes of non-response, and modifying immunosuppression to improve the response to the vaccines [69]. Several studies had shown that vaccines are not always offered to transplant patients [70] and that responses to several vaccines, such as influenza or hepatitis B, were not very high [69]. Risk factors for non-response to these vaccines were also identified. However, few studies have assessed the effect of modifying immunosuppression in this setting. Hence, the large number of studies performed with anti-SARS-CoV-2 vaccines should or will prompt transplant physicians to improve their vaccination policy in SOT patients. Thirdly, the technology of mRNA -based vaccination is also of great interest and will probably be used to develop other vaccines, such as those against HIV or other viruses. For instance, in dialysis patients who are candidates for kidney transplantation, the response to the anti-SARS-CoV-2 vaccine was excellent, while in the same population, it is difficult to obtain a humoral response after vaccination against the hepatitis B virus. New technology could, therefore, be developed to improve the virological response.

### 5.2. Anti-SARS-CoV2 Neutralizing Monoclonal Antibodies

Neutralizing monoclonal antibodies were rapidly developed and used to prevent SARS-CoV-2 infection in weak and non-responding immunosuppressed patients, to whom three vaccine doses were administered. Several groups, including ours, have shown that a monthly dose of casirivimab/imdevimab was highly effective in preventing Delta variant infection [44,71,72]. In a large cohort, no patient developed Delta variant infection, but this combination is not effective against the Omicron variant, both in vitro [73,74] and in vivo [75]. Currently, Evushled^®^, a combination of two monoclonal antibodies, Cilgavimab and Tixagevimab, is approved in several countries in this context, and they have recently demonstrated their efficacy in preventing COVID-19 infection [76]. More recently, due to breakthroughs observed in patients given pre-exposition Evushled^®^, it was recommended that the doses be doubled. In addition, the increased neutralizing activity of AZD7442 against the BA.2 subvariant will probably improve its efficacy in immunocompromised patients.

## 6. Conclusions

The COVID-19 pandemic has dramatically modified the management of SOT patients. Lessons from actions taken during the pandemic can be drawn on in order to modify and likely improve the management of such patients.

## Data Availability

Not applicable.

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
