# Peer review of "Can the COVID-19 Pandemic Improve the Management of Solid Organ Transplant Recipients?"

_viruses, 2022, doi:10.3390/v14091860_

Round 1
Reviewer 1 Report
Thank you for allowing me to review your paper entitled "Can the COVID-19 pandemic improve the management of solid organ transplant recipients?" This paper reviews the current data about COVID-19 in transplant patients. While the paper is interesting it is not what I was expecting from the title. To align with the title, I would add more discussion about how what we did or learned about during the COVID-19 pandemic could be applied to future pandemics. For example, we had MAbs but limited data on their utility inour population. Vaccine was approved without an appropriate set of studies in the population expected to have poor responses.
Author Response
Thank you for allowing me to review your paper entitled "Can the COVID-19 pandemic improve the management of solid organ transplant recipients?" This paper reviews the current data about COVID-19 in transplant patients.
While the paper is interesting it is not what I was expecting from the title.
To align with the title, I would add more discussion about how what we did or learned about during the COVID-19 pandemic could be applied to future pandemics. For example, we had MAbs but limited data on their utility in our population.
We thank the reviewer for his comments. We have added that both monoclonal antibodies and convalescent plasma therapies were successfully used in transplant patients without having preliminary data in this setting. This suggests that, in case of another pandemic, new therapeutics should be rapidly tested in SOT patients and could be used in this specific population.
Vaccine was approved without an appropriate set of studies in the population expected to have poor responses.
This was also added

Reviewer 2 Report
Your review titled “Can the Covid-19 pandemic improve the management of solid organ transplant recipients” certainly touches a very interesting topic. However, it seems to me that it should have been approached with more scientific depth, greater critical thinking, and eventually lead us to more thoughtful conclusions. For example, some parts as the “Virology laboratory expertise” are a mere presentation of what is very well known for SARS-CoV-2 molecular and serological assays in general. One cannot see the impact on SOT patients. The part “Management of immunosuppression during severe infectious disease complications” definitely needs some corrections, additions, and a clearer conclusion. What do you mean by saying “no acute rejection of de novo DSAs was observed”?. The phrasing is not right. Do they mean humoral rejection? Do you mean humoral rejection associated with the appearance of de novo DSAs? Then, “In regard to out-patients, the management of immunosuppression varied between centers”. This phrase has to expand and become a more informative and descriptive paragraph based on the literature you have reviewed.
I believe that if your review is given more extensive scientific thinking and mental labor, it will be an interesting paper to publish.
Author Response
Your review titled “Can the Covid-19 pandemic improve the management of solid organ transplant recipients” certainly touches a very interesting topic. However, it seems to me that it should have been approached with more scientific depth, greater critical thinking, and eventually lead us to more thoughtful conclusions. For example, some parts as the “Virology laboratory expertise” are a mere presentation of what is very well known for SARS-CoV-2 molecular and serological assays in general. One cannot see the impact on SOT patients.
As requested, the virological part was modified. We have focused on the impact of virological monitoring in SOT patients who have a delayed clearance of the virus. For example, the viral sequencing allowed to detected the emergence of viral mutations in SOT patients given strovimab or tixagevimab/cilgavimab.
The part “Management of immunosuppression during severe infectious disease complications” definitely needs some corrections, additions, and a clearer conclusion. What do you mean by saying “no acute rejection of de novo DSAs was observed”?. The phrasing is not right. Do they mean humoral rejection? Do you mean humoral rejection associated with the appearance of de novo DSAs?
We thank the reviewer for having detected a typo. The right sentence is ““no acute rejection OR (instead of OF) de novo DSAs was observed”. This was corrected.
Then, “In regard to out-patients, the management of immunosuppression varied between centers”. This phrase has to expand and become a more informative and descriptive paragraph based on the literature you have reviewed.
There are no robust published data regarding the management of immunosuppression in this setting. This was added in the manuscript. We have also added that in our center, immunosuppression was not modified in patients non-requiring hospitalization.
I believe that if your review is given more extensive scientific thinking and mental labor, it will be an interesting paper to publish.
We thank the reviewer for his comment. We have identified several lessons that can be used in SOT patients in the absence of the context of COVID (Isolation in the ward, wear of masks, immunosuppressants infusion at home, etc…) and others that can be used in case of another pandemic such as testing new therapeutics in this specific population.

Round 2
Reviewer 2 Report
agree